# Association between Serum Vitamin A, Blood Lipid Level and Dyslipidemia among Chinese Children and Adolescents

**DOI:** 10.3390/nu14071444

**Published:** 2022-03-30

**Authors:** Lianlong Yu, Yongjun Wang, Dongmei Yu, Shixiu Zhang, Fengjia Zheng, Ning Ding, Lichao Zhu, Qianrang Zhu, Wenkui Sun, Suyun Li, Gaohui Zhang, Liangxia Chen, Yiya Liu, Li Yang, Jian Feng

**Affiliations:** 1Shandong Center for Disease Control and Prevention, Jinan 250014, China; lianlong00a@163.com (L.Y.); zhengfengjia1985@163.com (F.Z.); wenkuisun@163.com (W.S.); lsyjkzx@163.com (S.L.); gaohui1221@sina.com (G.Z.); clx1617@163.com (L.C.); 2Department of Clinical Nutrition, The First Affiliated Hospital of Shandong First Medical University & Shandong Provincial Qianfoshan Hospital, Jinan 250014, China; wangyongjun519@163.com; 3National Institute for Nutrition and Health, Chinese Center for Disease Control and Prevention, Beijing 100050, China; yudm@ninh.chinacdc.cn; 4Department of Nutrition and Food Hygiene, School of Public Health, Cheeloo College of Medicine, Shandong University, Jinan 250012, China; amyzhangsx@sdu.edu.cn; 5Department of Pediatrics, Shandong Provincial Hospital Affiliated to Shandong First Medical University, Jinan 250021, China; dingning8903@163.com; 6Department of Pediatric Surgery, Shandong Provincial Hospital Affiliated to Shandong First Medical University, Jinan 250021, China; zhu-lichao0217@163.com; 7Jiangsu Provincial Center for Disease Control and Prevention, Nanjing 210009, China; zhuqianrang@hotmail.com; 8Guizhou Center for Disease Control and Prevention, Guiyang 550001, China; liuyiya163@163.com; 9Jinan Center for Disease Control and Prevention, Jinan 250021, China; 10Department of Clinical Nutrition, Qilu Hospital of Shandong University, Jinan 250012, China

**Keywords:** vitamin A, lipids, dyslipidemia, children, adolescents

## Abstract

Background: To study the relationship between serum vitamin A (VA) level and blood lipid profiles in children and adolescents aged 6–18 years, as well as the effect of VA on dyslipidemia. Methods: The project adopted a multistage stratified cluster sampling method. The Food Frequency Questionnaire (FFQ) was used to obtain dietary factors data. Blood samples of subjects were taken via venipuncture. Generalized linear models were used to explore the correlation be-tween VA and biochemical indicators, as well as stratified and inter-actions analysis to explore the influence of confounders on these relationships. Generalized linear models were constructed to explore the association between VA and blood lipids. Restricted cubic splines were used to characterize dose–response associations between serum VA and dyslipidemia based on logistic regression. Results: Serum VA was positively correlated with TC, TG and HDL-C (*p* < 0.05), but these associations were influenced by age (*p* < 0.05). The adjusted odds ratio (OR) values of VA for hypercho lesterolemia, hypertriglyceridemia, mixed hyperlipidemia and low high-density lipoprotein cholesterolemia were 3.283, 3.239, 5.219 and 0.346, respectively (*p* < 0.01). Meanwhile, significant age interactions affected the relationship between VA and TC, as well as TG and LDL-C (*p* < 0.01). Conclusion: Serum VA was positively correlated with blood lipids, but these associations were influenced by age. VA was a risk factor for dyslipidemias, such as hypercholesterolemia, hypertriglyceridemia and mixed hyperlipidemia, but was a protective factor for low high-density lipoprotein cholesterolemia.

## 1. Introduction

As a kind of fat-soluble vitamin, vitamin A (VA) is one of the most essential nutrients for maintaining the normal growth and development of children and is closely related to children’s health [1]. According to the WHO recommendation, children’s serum retinol concentration ≥ 1.05 μmol/L can be determined as vitamin adequacy [2,3]. In clinical work, we found that the over-use of VA can lead to dyslipidemia. However, few studies have focused on the correlativity between serum VA and blood lipids, as well as between serum VA and dyslipidemia in children and adolescents. For cardiovascular diseases (CVDs), such as coro-nary artery disease and stroke, dyslipidemia is a considerable risk factor [4]. Dyslipidemia, if developed in childhood, is associ-ated with an increased risk of developing atherosclerosis later in life. The incidence of dyslipidemia is as high as 20.6–31.6% among children and adolescents in China [5,6], close to the prev-alence of adults, which suggests a potentially large burden in terms of blood lipids and CVDs in adulthood [7].

Previous studies showed that nutritional factors have im-portant influences on dyslipidemia [8]. VA is known to play a very important role in lipid metabolism [9,10,11]. The level of VA and its metabolite retinoic acid (RA) regulate the pivotal lipogen-ic enzymes, as well as their gene expressions involved in lipid metabolism [12]. Epidemiological and mechanism studies have indicated that VA can stimulate lipid catabolism in some tissues and contribute to reducing the incidence of obesity [13] and the risk of atherosclerotic cardiovascular disease (ASCVD) [14,15,16]. However, a contradictory association between VA and plasma lipid metabolism was observed in long-term retinoid drug users and people who took excessive amounts of VA supplements with symptoms of hypercholesterolemia, hypertriglyceridemia and high serum low-density lipoprotein levels [17]. The medical ad-ministration of isotretinoin (13-cis-RA) can lead to elevated blood triglycerides in patients with acne [18]. Similarly, some studies found that taking all-trans RA in patients with acute promyelo-cytic leukemia increased plasma TG and cholesterol levels [19,20]. In addition to clinical observation, animal experimental studies also demonstrated that plasma HDL-C levels decreased in hypercholesterolemic obese rats fed with a vitamin-A-enriched diet with an effective dose of 52 mg/kg [21]. Contrary to the above research results, preclinical studies indicated that VA deficiency increased the levels of triglyceride and circulating cholesterol [22,23]. Animal studies found that the levels of serum triglycer-ide, cholesterol and high-density lipoprotein decreased in VA-deficient rats, as well as the phospholipid content in the liver [24]. The effect of VA deficiency in rodents may be caused by the decreased activity of fatty acid synthesis in the liver and the im-paired synthesis of mevalonate cholesterol. Two more animal studies separately showed that VA deficiency prevented high-fructose diet-induced TAG in the liver and plasma in rats [25] and high-fat diet-induced steatosis in mice [26]. Therefore, the level of serum VA may have different health effect magni-tudes than lipid metabolism, and the findings are not clear enough based on previous studies regarding the relationship between VA deficiency and circulating lipoproteins in humans.

Given that VA plays a complex role in lipid metabolism, further research should be done on lipid metabolism in response to the VA level. At the same time, we need to explore the rela-tionship between children and adolescents of different ages in order to provide effective health intervention strategies to reduce the risk of dyslipidemia and improve cardiovascular health. This study explored the correlations between serum VA level, lipid profile and dyslipidemia among children and adolescents aged 6–18 years. This study also explored the role of age in these associations.

## 2. Materials and Methods

### 2.1. Study Design

This study was based on the survey of “China Nutrition and Health Surveillance (2016–2017)”, which was approved by the Chinese Center for Disease Control and Prevention (China CDC). The project adopted a multi-stage stratified cluster sampling method and selected 13 districts and counties in Shandong Province, namely, Yishui, Shibei, Lingcheng, Lijin, Shouguang, Penglai, Wucheng, Lanshan, Dong’e, Laizhou, Linzi, Dingtao, and Sishui. A total of 26 primary schools, 26 junior high schools, and 13 high schools were included in this survey. A total of 3551 adolescents were incorporated into the study. Inclusion and exclusion criteria: At least 280 children and adolescents aged 6–17 years needed to be surveyed at each monitoring site, including ten classes in primary school grades 1–6, junior one, junior two, senior one and senior two. There were 28 students in one class, half male and half female. Subjects with obvious diseases were excluded. During the investigation process, tablet and computer programs were used to investigate quality control, and subjects who were unable to participate were promptly replaced.

This survey was approved by the Ethics Committee of the National Institute for Nutrition and Health and the Chinese Centre for Disease Control and Prevention (number: 201614), and the participants understood all aspects of the informed consent and attended this survey voluntarily.

A questionnaire survey, body examination, dietary interview and laboratory test were included in this research. The working groups from the nation and provincial working groups were responsible for the quality control of the survey. The staff at the district and county levels, who were uniformly trained by the local Center for Disease Control and Prevention, were responsible for collecting fasting blood samples and undertaking the interviews and questionnaires.

### 2.2. Sample Collection Methods

All subjects underwent a physical examination, including routine basic items, such as blood pressure, weight, waist circumference and height measurements. Blood samples of subjects were taken via venipuncture. Serum and plasma samples were separated using the standard rules of the working group and were stored at −86 °C, then sent to the China CDC for a unified quantitative analysis in the laboratory through the cryogenic cold chain.

The Food Frequency Questionnaire (FFQ) [27], designed by the China CDC, was used to obtain dietary factors data. Young participants under 12 years old completed the survey with the help of their guardians.

The participants were classified into three groups based on WHO criteria [2,3] and Chinese pediatric textbooks [28]: VA deficiency (<0.7 μmol/L), edge VA deficiency (0.7–1.05 μmol/L) and normal (≥1.05 μmol/L).

Meanwhile, according to the expert consensus on the prevention and treatment of dyslipidemia among Chinese children and adolescents, dyslipidemias was defined as hypercholesterolemia (TC ≥ 5.18 mmol/L), hypertriglyceridemia (TG ≥ 1.70 mmol/L), mixed hyperlipidemia (TC ≥ 5.18 mmol/L and TG ≥ 1.70 mmol/L) and low high-density lipoprotein cholesterolemia (HDL-C ≤ 1.04 mmol/L) [29].

### 2.3. Statistical Methods

Statistical analysis was performed using one-way ANOVA and post hoc comparisons involved the SNK test for continuous variables to compare different VA level groups in terms of biochemical indicators and demographic characteristics and the chi-square test for categorical variables. Then, the generalized linear models were used to explore the correlation between VA and biochemical indicators, as well as employed stratified and interactions analysis to explore the influence of confounders on these relationships.

Logistic regression was used to calculate 95% confidence intervals and odds ratios (ORs). Generalized linear models were constructed to explore the association between VA and blood lipids. Model 1 was adjusted for sex (male/female). Model 2 was further adjusted for hs-CRP intake (continuous), fat intake (continuous); systolic pressure (continuous), diastolic pressure (continuous), BMI (continuous), hemoglobin (continuous), blood glucose (continuous), blood uric acid (continuous), serum creatinine (continuous), ferritin (continuous), transferrin receptor (continuous), albumin (continuous), total protein (continuous), serum Zn (continuous) and vitamin D (continuous). Model 3 was adjusted for age (continuous) and sex (male/female). Model 4 was further adjusted for hs-CRP intake (continuous), fat intake (continuous), systolic pressure (continuous), diastolic pressure (continuous), BMI (continuous), hemoglobin (continuous), blood glucose (continuous), blood uric acid (continuous), serum creatinine (continuous), ferritin (continuous), transferrin receptor (continuous), albumin (continuous), total protein (continuous), serum Zn (continuous) and vitamin D (continuous). Logistic regression analysis was used to evaluate the relationship between dyslipidemias and risk factors. The dependent variable in this model was whether the participants had different kinds of dyslipidemia (hypercholesterolemia, hypertriglyceridemia, mixed hyperlipidemia and low high-density lipoprotein cholesterolemia). Restricted cubic splines were used to characterize dose–response associations between serum VA and dyslipidemia based on logistic regression. Residual confounding was also minimized for confounders using restricted cubic splines (smooth curves). R version 3.6.3 was used to generate graphs. Any *p* < 0.05 was considered statistically significant.

## 3. Results

### 3.1. Description of the Sample Characteristics

The average age of the participants was 11.4 years and 49.7% were males among the 3551 participants in the study. Meanwhile, 63.8% of the subjects were aged 6–12 years and 36.2% were aged 13–18 years. There were 1.6% of the subjects with serum VA deficiency status (<0.7 μmol/L), 22.2% with edge VA deficiency status (0.7–1.05 μmol/L) and 76.2% with normal VA status (>1.05 μmol/L). As shown in Table 1, there were no significant differences between different VA status groups regarding protein, fat and energy intakes (*p* > 0.05). Subjects with a lower level of VA deficiency status had a significantly younger age (*p* < 0.01).

### 3.2. VA and Blood Lipid Concentrations

Figure 1 shows the association of age with serum VA. It was found that the serum VA concentration of the subjects increased with age, and the change curve was almost the same for males and females. Figure 2 shows the association of age with blood lipids, showing that a significant non-linear association was found between age and TC, TG, HDL-C and LDL-C. Subjects with a lower level of VA status had significantly lower systolic pressure, weight, height, lower hemoglobin, blood glucose, total cholesterol, blood uric acid, serum creatinine, albumin and vitamin D (*p* < 0.01) but higher hs-CRP (*p* < 0.01) when compared to the subjects in the serum vitamin A normal group (Table 1).

### 3.3. Distribution of Blood Lipids in Different Vitamin A Status

As depicted in Table 2, subjects that were 6–12 years old with a higher level of VA status had significantly higher TC, TG, HDL-C and LDL-C (*p* < 0.0001). In the group of 13–18-year-olds, subjects with a higher level of VA status had significantly higher TC, TG and HDL-C (*p* < 0.001) but not LDL-C (F = 2.49, *p* = 0.083).

### 3.4. Description of Dyslipidemias and Risk Factors

The results of the logistic regression analysis are shown in Table 3. The odds ratio (OR) values of VA for hypercholesterolemia, hypertriglyceridemia, mixed hyperlipidemia and low high-density lipoprotein cholesterolemia were 3.3, 3.2, 5.2 and 0.3, respectively (*p* < 0.01). Furthermore, hypercholesterolemia was negatively correlated with age (OR = 0.84, *p* < 0.0001) and positively correlated with vitamin D (OR = 1.03, *p* = 0.005), ferritin (OR = 1.004, *p* = 0.042) and total protein (OR = 1.09, *p* < 0.0001). Hypertriglyceridemia was negatively associated with vitamin D (OR = 0.97, *p* = 0.008) and serum creatinine (OR = 0.97, *p* = 0.001), and positively correlated with BMI (OR = 1.03, *p* < 0.0001), blood uric acid (OR = 1.005, *p* < 0.001), transferrin receptor (OR = 1.20, *p* < 0.0001), albumin (OR = 1.09, *p* = 0.015) and serum zinc (OR = 1.012, *p* < 0.0001). Mixed hyperlipidemia was negatively correlated with serum creatinine (OR = 0.94, *p* = 0.021) and positively correlated with BMI (OR = 1.027, *p* = 0.032). Low HDL cholesterolemia was negatively correlated with albumin (OR = 0.86, *p* < 0.0001) and positively correlated with BMI (OR = 1.026, *p* = 0.001), serum uric acid (OR = 1.005, *p* < 0.0001), ferritin (OR = 1.009, *p* < 0.0001) and transferrin receptor (OR = 1.16, *p* = 0.004).

### 3.5. Description of Multi-Variable Associations of Vitamin A with Blood Lipids

The generalized linear model (Table 4) showed that after adjustment for gender, age, hs-CRP, protein intake, fat intake, systolic pressure, diastolic blood pressure, BMI, hemoglobin, blood glucose, blood uric acid, serum creatinine, ferritin, transferrin receptor, albumin, total protein, serum zinc and vitamin D, VA was significantly positively correlated with TC, TG, HDL-C and LDL-C (*p* < 0.0001). Meanwhile, a significant age interaction affected the relationship between VA and TC, TG and LDL-C (*p* < 0.01).

### 3.6. Description of Serum VA and Dyslipidemia

Figure 3 shows the associations of the ORs of dyslipidemias with the increase in serum VA concentration. With the increase in serum VA concentration, the effect of VA switched from protective to pathogenic for hypercholesterolemia and hypertriglyceridemia, but it showed the opposite trend for low high-density lipoprotein cholesterolemia.

## 4. Discussion

The results revealed that serum VA was positively correlated with TC, TG, LDL-C and HDL-C levels, and these associations were influenced by age. Serum VA levels were positively correlated with TC, TG, HDL-C and LDL-C in the 6–12-year-olds group, while in the 13–18-year-olds group, serum VA levels were significantly positively correlated with TC, TG and HDL-C. It was shown that the metabolism and storage of VA during some disease progression is related to cholesterol and triglyceride [6]. Animal studies showed that VA deficiency can result in a hypolipidemic effect by lowering serum TC, TG and HDL-C levels in male rats for 21 days after being fed a VA-deficient diet for 3 months [30]. The study results were consistent with the results produced here. In fact, few population investigations or clinical trials have reported the relationship between serum VA and blood lipids. However, retinoids and VA derivatives are widely used as effective routine drugs in the treatment of skin diseases, playing a role in proliferation inhibition, differentiation induction and anti-inflammatory or lipid inhibition. In addition, VA derivatives were also clinically observed to affect lipid metabolism, which is mainly manifested as an increase in triglyceride or cholesterol levels [31].

Furthermore, VA was a risk factor for dyslipidemias, namely, hypercholesterolemia, hypertriglyceridemia and mixed hyperlipidemia, but was a protective factor for low high-density lipoprotein cholesterolemia. Studies showed that the use of a retinoid could cause hyperlipidemia in the treatment of dermatology diseases, where the incidence of hyperlipidemia was more than 30% [32]. This also confirmed the positive correlation between VA and blood lipids. Prior studies noted that VA and/or VA-related parameters (such as VA-binding proteins and enzymes) played important and different kinds of roles in the development of metabolic disease [11]. The results of this study indicated that when the concentration increased, the effect of VA switched from protective to pathogenic for hypercholesterolemia and hypertriglyceridemia; however, the opposite trend was found for low high-density lipoprotein cholesterolemia. Therefore, it is important to find the upper limit of serum VA concentration.

In general, the physiological effects of VA were mediated by its metabolite retinoic acid (RA). Various subtypes of RA control gene expression by regulating the activity of RARs and RXRs [33,34,35,36,37] to affect key lipogenic enzymes and their genes. Retinoids can upregulate apolipoprotein-CIII (APO-CIII) [38]. Lipoproteins, such as VLDL and chylomicron, are rich in triglycerides and produce APO-CIII, which acts as a noncompetitive inhibitor of lipoprotein lipase (LPL). This provides one explanation for the elevated triglyceride levels in patients taking medicated retinoids [13]. Another possible explanation is that retinoid upregulates the reactivity of fatty acid synthase and thus increases liver fatty acid synthesis, although increased liver fatty acid oxidation may compensate for some of these effects [39]. As mentioned in the previous studies, the VA signaling system regulates acetyl-CoA carboxylase (ACC) activity and expression. The synthesis and composition of lipids are altered by VA deficiency; meanwhile, VA deficiency can significantly reduce ACC activity [40,41]. Most studies showed that sufficient carotenoids and VA protect against atherosclerosis, while the excess or lack of VA increases the risk of arteriosclerotic cardiovascular disease [42]. Hence, the overall effect of VA on hyperlipidemia may exist within a concentration range. In short, dyslipidemia in animals, whether caused by VA deficiency or VA excess, was a complex multi-causal effect [43]. The different responses of lipid metabolism to VA levels deserve further study. In order to reveal the effects of hyperlipidemia on VA metabolism, as well as the role of VA in the development and treatment of hyperlipidemia, more carefully designed clinical and laboratory studies are needed.

The study results also showed that there were 1.6% of the subjects with VA deficiency (<0.7 μmol/L) and 22.2% with edge VA deficiency (0.7–1.05 μmol/L). The overall VA deficiency rate was 23.8%, i.e., almost a quarter of surveyed children present serum VA deficiency. On the basis of the results of nutrition and health monitoring of Chinese residents (2010–2012), the prevalence of VA deficiency in children in China was 7.69% in urban areas and 5.53% in rural areas [44,45]; this indicates the prevalence of VA inadequacy in children in China is still at a high level, which is consistent with other most emerging developing countries. The prevalence of VA inadequacy in children in China was 18.57% in urban areas and 18.07% in rural areas [44,45]. The relationship between different VA plasma levels and sex, age, dietary factors, anthropometric data and biochemical indicators was performed. The results demonstrated that the weight and height of subjects with VA deficiency were lower than those in the edge deficiency and normal groups. VA deficiency reduces immune capacity and leads to increased morbidity and mortality due to night blindness, corneal ulceration, keratomalacia, xerophthalmia and related ocular symptoms [46]. It also increased mortality due to measles, diarrhea and respiratory diseases [42,47,48,49,50,51,52]. Therefore, more attention to the problem of VA deficiency in children and adolescents should be paid.

In the current study, correlation analyses provided evidence that higher inflammatory parameters, such as C-reactive protein, were significantly associated with reduced plasma VA levels. This finding supported available information showing that reduced plasma levels of VA were often associated with infection and acute inflammatory conditions [10,12,53,54].

The study had some limitations. First, the methods and results data were from a cross-sectional health examination survey. This study can only reveal correlative relationships, not causality. Further studies are needed to combine the effects of VA on mRNA and protein expression of critical enzymes of fatty acid synthesis in vivo and in vitro to find the causal relationship between VA level and blood lipids or the effect of dyslipidemia on the plasma VA level. Second, the study only observed participants aged 6–18 years, which might not reflect the relationship between VA and lipids in the whole population. In view of the above limitations, further prospective, large-sample cohort studies are needed.

Despite the above limitations, the present study had several notable strengths. The study found that there was a close correlation between VA level and blood lipids. Very little was found in the literature on the association between serum VA and blood lipids in a large sample population, especially in children and adolescents, and this study has clarified this issue.

## 5. Conclusions

In conclusion, serum VA was positively correlated with TC, as well as TG and HDL-C, but these associations were influenced by age. Serum VA levels were significantly positively correlated with TC, TG, HDL-C and LDL-C in the 6–12-year-olds group, while in the 13–18-year-olds group, serum VA levels were significantly positively correlated with TC, TG and HDL-C (*p* < 0.05). VA was a risk factor for dyslipidemias, namely, hypercholesterolemia, hypertriglyceridemia and mixed hyperlipidemia, but was a protective factor for low high-density lipoprotein cholesterolemia. These results will provide a reference for the discovery of pathogenesis in pediatric hyperlipidemia.

## Figures and Tables

**Figure 1 nutrients-14-01444-f001:**
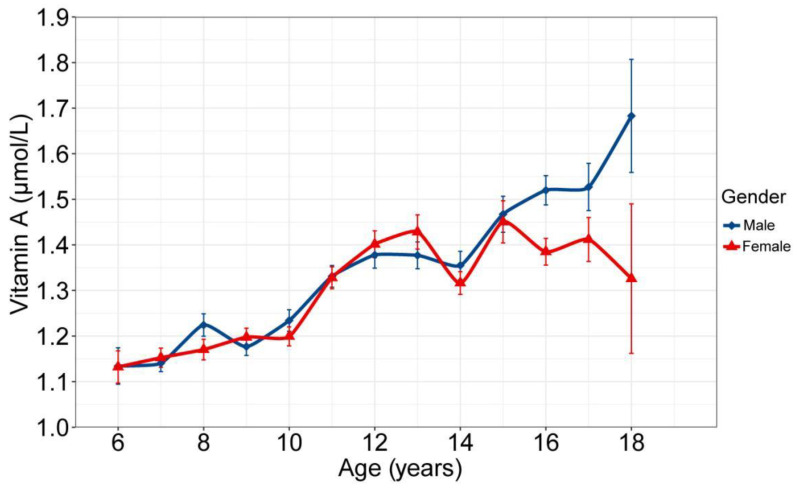
Change in vitamin A with age for different sexes. Means and standard error of the means are shown.

**Figure 2 nutrients-14-01444-f002:**
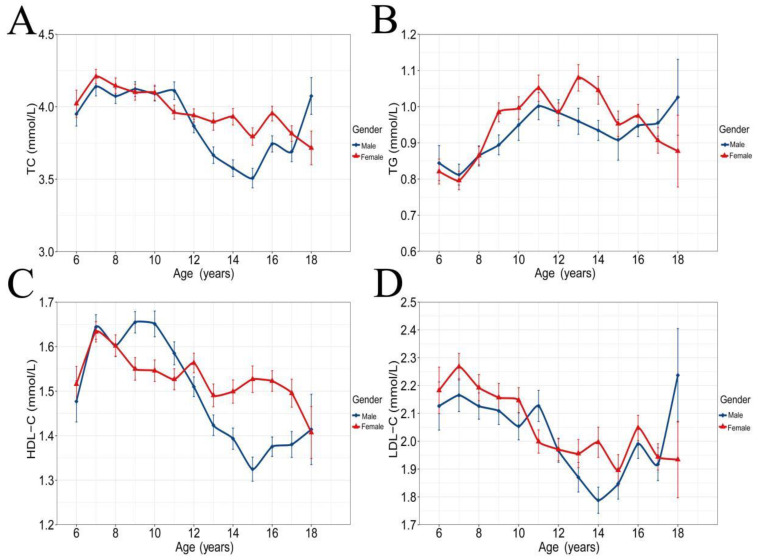
Change in blood lipids with age for different sexes. Means and standard error of the means are shown. (**A**) Change in TC with age. (**B**) Change in TG with age. (**C**) Change in HDL-C with age. (**D**) Change in LDL-C with age.

**Figure 3 nutrients-14-01444-f003:**
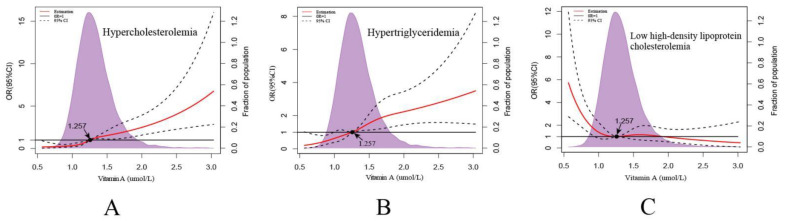
OR and 95%CI for dyslipidemia from logistic regression using restricted cubic splines. Analyses were adjusted for age and sex. Solid red lines are multivariable adjusted odds ratios, with dashed blues lines showing 95% confidence intervals derived from restricted cubic spline regressions with three knots.
(**A**) RCS logistic regression models for vitamin A and ORs of hypercholesterolemia.
(**B**) RCS logistic regression models for vitamin A and ORs of hypertriglyceridemia. (**C**) RCS logistic regression models for vitamin A and ORs of low high-density lipoprotein cholesterolemia.

**Table 1 nutrients-14-01444-t001:** Sample characteristics according to vitamin A status (mean values and standard deviations; numbers and percentages).

Variables	Total (N = 3551, %)		Vitamin A Deficiency (<0.7 μmol/L)		Edge Vitamin A Deficiency (0.7–1.05 μmol/L)		Normal (≥1.05 μmol/L)		χ^2^/F	*p*
	Mean	SD	Mean	SD	Mean	SD	Mean	SD		
Total (n, %)	3551 (100.0)		58 (1.6)		786 (22.2)		2707 (76.2)			
Male (n, %)	1766 (49.7)		32 (0.9)		361 (10.3)		1373 (38.6)		6.3	0.043
Female (n, %)	1785 (50.2)		26 (0.7)		425 (11.9)		1334 (37.6)			
Age (years)	11.4	3.2	10.1	3	10.4	3.1	11.7	3.1	59.1	<0.001
6–12 years (n, %)	2265 (63.8)		47 (1.3)		585 (16.5)		1633 (46.0)		60.0	<0.001
13–18 years (n, %)	1286 (36.2)		11 (0.3)		201 (5.7)		1074 (30.2)			
Dietary factors										
Protein intake (g/day)	139.5	131.6	139.1	141.4	140.8	185	139.2	111.3	0.04	0.959
Fat intake (g/day)	52.1	76.3	50.5	55.9	50.4	74.8	52.7	77.1	0.3	0.751
Energy intake (g/day)	2388.9	2104.4	2301.9	2052.8	2348.7	2329.3	2402.5	2036.2	0.2	0.780
Anthropometrics										
Systolic pressure (mmHg)	114.1	12.4	110.5	11.4	111	11.9	115.1	12.4	35.1	<0.001
Diastolic pressure (mmHg)	67.9	9.2	67.7	7.6	66.9	9.1	68.2	9.1	6.2	0.002
Weight (kg)	45.2	16.5	37.8	14.5	38.5	14	47.2	16.7	95.0	<0.001
Height (cm)	150.5	16.7	142.9	16.6	144.8	16.6	152.4	16.3	72.9	<0.001
BMI (kg/m^2^)	19.4	5.9	17.9	3.4	17.8	3.4	19.9	6.4	40.4	<0.001
Biochemistry										
Hemoglobin (g/L)	136.8	13.3	130.8	13.6	131.6	11.7	138.5	13.3	92.7	<0.001
Blood glucose (mmol/L)	5.2	0.4	5.1	0.6	5.2	0.4	5.3	0.4	15.6	<0.001
Total cholesterol (mmol/L)	3.9	0.7	3.7	0.8	3.8	0.6	4.0	0.7	46.0	<0.001
Triglyceride (mmol/L)	0.9	0.4	0.8	0.3	0.8	0.3	1	0.4	38.3	<0.001
HDL-C (mmol/L)	1.5	0.3	1.5	0.4	1.5	0.3	1.5	0.3	6.4	0.002
LDL-C (mmol/L)	2.0	0.6	1.9	0.8	1.9	0.5	2.1	0.6	18.3	<0.001
Blood uric acid (μmol/L)	316.6	82.6	279.7	76.7	284.6	69.1	326.6	83.7	89.0	<0.001
Serum creatinine (μmol/L)	52.7	13.2	47.3	11.2	47.4	10.7	54.3	13.5	94.3	<0.001
Ferritin (ng/mL)	62.0	39.2	69.0	48.6	55.7	31.5	63.7	40.8	13.5	<0.001
Transferrin receptor (mg/L)	3.2	1.1	3.3	1.0	3.3	1.2	3.2	1.1	3.5	<0.001
hs-CRP (mg/L)	1.0	3.2	2.7	5.3	1.4	4.3	0.8	2.6	18.3	<0.001
Albumin (g/L)	49.5	2.9	47.5	2.7	48.3	2.6	49.9	2.9	107.9	<0.001
Total protein (g/L)	76.5	4.5	75.4	4.6	75.0	4.4	76.9	4.5	57.5	<0.001
Serum Zn (μmol/L)	87.5	18.1	87.4	24.7	84.9	18.3	88.2	17.8	9.9	<0.001
Vitamin D (ng/mL)	17.6	6.6	13.9	5.8	16.7	6.2	17.9	6.7	17.8	<0.001

**Table 2 nutrients-14-01444-t002:** Blood lipid values according to age group and vitamin A status (mean values and standard deviations).

Variables	Vitamin A Deficiency (<0.7 μmol/L		Edge Vitamin A Deficiency (0.7–1.05 μmol/L)		Normal (≥1.05 μmol/L)		F	*p*
6–12 Years	Mean	SD	Mean	SD	Mean	SD		
TC (mmol/L)	3.7	0.9	3.8	0.6	4.1	0.7	58.9	<0.001
TG (mmol/L)	0.7	0.2	0.8	0.3	0.9	0.4	30.3	<0.001
HDL-C (mmol/L)	1.5	0.4	1.5	0.3	1.6	0.3	9.2	<0.001
LDL-C (mmol/L)	1.9	0.8	1.9	0.6	2.2	0.7	24.4	<0.001
13–18 years								
TC (mmol/L)	3.5	0.5	3.6	0.6	3.8	0.7	9.4	<0.001
TG (mmol/L)	0.8	0.4	0.9	0.4	0.9	0.4	5.9	0.003
HDL-C (mmol/L)	1.3	0.3	1.4	0.3	1.5	0.3	5.2	0.005
LDL-C (mmol/L)	1.9	0.4	1.9	0.5	2	0.6	2.5	0.083

SD: standard deviation.

**Table 3 nutrients-14-01444-t003:** Logistic regression analysis of the relationship between dyslipidemias and risk factors.

	Hypercholesterolemia				Hypertriglyceridemia				Mixed Hyperlipidemia				Low High-Density Lipoprotein Cholesterolemia			
	OR	95%CI		*p*	OR	95%CI		*p*	OR	95%CI		*p*	OR	95%CI		*p*
Age (years)	0.8	0.7	0.9	<0.001	1.0	0.9	1.1	0.908	1.0	0.8	1.2	0.778	1.1	1.0	1.2	0.193
Sex	1.1	0.8	1.5	0.605	0.9	0.7	1.3	0.599	1.1	0.5	2.4	0.871	0.9	0.6	1.4	0.782
BMI (kg/m^2^)	1.0	1.0	1.0	0.810	1.0	1.0	1.0	<0.001	1.0	1.0	1.1	0.032	1.0	1.0	1.0	0.001
Vitamin A (μmol/L)	3.3	2.2	5.0	<0.001	3.2	2.1	4.9	<0.001	5.2	2.2	12.4	0.000	0.3	0.2	0.6	0.000
Vitamin D (ng/mL)	1.0	1.0	1.1	0.005	1.0	0.9	1.0	0.008	1.0	0.9	1.1	0.735	1.0	0.9	1.0	0.101
Blood glucose (mmol/L)	1.0	0.7	1.5	0.850	1.0	0.7	1.5	0.948	1.5	0.8	2.6	0.166	0.7	0.5	1.1	0.156
Blood uric acid (μmol/L)	1.0	1.0	1.0	0.919	1.0	1.0	1.0	<0.001	1.0	1.0	1.0	0.054	1.0	1.0	1.0	0.000
Serum creatinine (μmol/L)	1.0	1.0	1.0	0.411	1.0	1.0	1.0	0.001	0.9	0.9	1.0	0.021	1.0	1.0	1.0	0.284
Ferritin (ng/mL)	1.0	1.0	1.0	0.042	1.0	1.0	1.0	0.249	1.0	1.0	1.0	0.504	1.0	1.0	1.0	0.000
Transferrin receptor (mg/L)	1.1	1.0	1.3	0.051	1.2	1.1	1.3	0.000	1.1	0.9	1.4	0.237	1.2	1.1	1.3	0.004
hs-CRP (mg/L)	1.0	0.9	1.0	0.534	1.0	1.0	1.1	0.849	1.0	0.9	1.1	0.572	1.0	1.0	1.0	0.990
Albumin (g/L)	1.0	0.9	1.1	0.928	1.1	1.0	1.2	0.015	1.1	1.0	1.4	0.128	0.9	0.8	0.9	0.000
Total protein (g/L)	1.1	1.0	1.1	<0.001	1.0	0.9	1.0	0.087	1.0	0.9	1.1	0.482	1.0	1.0	1.1	0.540
Dietary Factors																
Dietary energy (kcal/day)	1.0	1.0	1.0	0.100	1.0	1.0	1.0	0.935	1.0	0.9	1.1	0.650	1.0	1.0	1.0	0.935
Dietary carbohydrate (g/day)	1.0	1.0	1.0	0.668	1.0	0.9	1.1	0.838	1.1	0.7	1.7	0.670	1.0	0.9	1.1	0.838
Dietary fat (g/day)	1.0	1.0	1.0	0.286	1.0	0.8	1.3	0.956	1.3	0.4	4.6	0.668	1.0	0.8	1.3	0.956
Dietary protein (g/day)	1.0	1.0	1.0	0.091	1.0	0.9	1.0	0.288	1.1	0.9	1.4	0.447	1.0	0.9	1.0	0.288
Dietary total carotene (μg/day)	1.0	1.0	1.0	0.904	1.0	1.0	1.0	0.623	1.0	0.9	1.1	0.745	1.0	1.0	1.0	0.623
Dietary retinol (μg/day)	1.0	1.0	1.0	0.242	1.0	1.0	1.0	0.618	1.0	0.7	1.6	0.862	1.0	1.0	1.0	0.618
Dietary vitamin A (μg/day)	1.0	1.0	1.0	0.774	1.0	1.0	1.0	0.626	0.9	0.6	1.4	0.745	1.0	1.0	1.0	0.626
Dietary vitamin E (mg/day)	1.0	1.0	1.0	0.927	1.0	0.9	1.0	0.321	0.9	0.5	1.9	0.823	1.0	0.9	1.0	0.321

**Table 4 nutrients-14-01444-t004:** Multi-variable associations of vitamin A with Blood lipid in children in Shandong province.

		Model 1			Model 2			Model 3			Model 4		*p*-Value Interaction for Age	*p*-Value Interaction for BMI
Blood lipids	Coefficient	95%CI	*p*	Coefficient	95%CI	*p*	Coefficient	95%CI	*p*	Coefficient	95%CI	*p*		
TC (mmol/L)	0.319	0.257 0.381	<0.001	0.315	0.245 0.385	<0.001	0.473	0.410 0.536	<0.001	0.347	0.278 0.418	<0.001	<0.001	0.462
TG (mmol/L)	0.242	0.206 0.279	<0.001	0.229	0.188 0.272	<0.001	0.232	0.194 0.269	<0.001	0.229	0.187 0.272	<0.001	0.001	0.478
HDL-C (mmol/L)	−0.0003	(−0.029 0.029)	0.998	0.075	0.043 0.108	<0.001	0.0572	0.027 0.087	<0.001	0.083	0.051 0.116	<0.001	0.425	0.946
LDL-C (mmol/L)	0.21	0.154 0.267	<0.001	0.134	0.070 0.198	<0.001	0.311	0.254 0.369	<0.001	0.158	0.094 0.222	<0.001	0.016	0.747

Model 1: adjusted for sex (male/female). Model 2: Further adjusted for hs-CRP intake (continuous), fat intake (continuous), systolic pressure (continuous), diastolic pressure (continuous), BMI (continuous), hemoglobin (continuous), blood glucose (continuous), blood uric acid (continuous), serum creatinine (continuous), ferritin (continuous), transferrin receptor (continuous), albumin (continuous), total protein (continuous), serum Zn (continuous) and vitamin D (continuous). Model 3: adjusted for age (continuous) and sex (male/female). Model 4: Further adjusted for hs-CRP intake (continuous), fat intake (continuous), systolic pressure (continuous), diastolic pressure (continuous), BMI (continuous), hemoglobin (continuous), blood glucose (continuous), blood uric acid (continuous), serum creatinine (continuous), ferritin (continuous), transferrin receptor (continuous), albumin (continuous), total protein (continuous), serum Zn (continuous) and vitamin D (continuous).

## Data Availability

The data are not allowed to be disclosed according to the National Institute for Nutrition and Health, Chinese Center for Disease Control and Prevention.

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
