# Peer review of "Association between Serum Vitamin A, Blood Lipid Level and Dyslipidemia among Chinese Children and Adolescents"

_nutrients, 2022, doi:10.3390/nu14071444_

Round 1

Reviewer 1 Report

I read with interest the research on the association between serum vitamin A, lipid status, and dyslipidaemia among Chinese children and adolescents from age 6 to 18 years. The study included a significant sample size of children and adolescents in a cross-sectional study. The authors during this study have shown that almost a quarter of the participants manifested vitamin A deficiency. They had shown a significant positive association between serum vitamin A with serum lipid parameters, i.e. triglycerides, total cholesterol and low HDL-cholesterol, which were significantly influenced by their age. Those results are important because they can serve to paediatric society for future health intervention strategies aiming to reduce the risk of dyslipidaemia and to protect cardiovascular health in adulthood. Still, the authors need to improve the manuscript in which they show their study' results and at the same time to clarify their statements. The presentation of results is uneven, deficient in some variables (dietary), and need to be improved with additional analyses which may explain the results and associations. I suggest doing additional analyses, especially in the dietary variables, to correct the section of the study's methods, to improve the presentations of their results, and to improve the English language editing of the whole manuscript.

Specific comments and suggestions are listed below.

Abstract

I suggest writing in passive voice and avoiding mentioning “we”. In methods mention only that statistical analysis/es that best describes the association between serum VA and blood lipids and dyslipidaemia. Avoid upper letters for Hypercholesterolemia, Hypertriglyceridemia, etc. I think that there is a mistake in the sentence that starts in line 34.

Introduction

Line 48: the sentence means that the authors found the statement in their clinical work or is this other research?

Line 51: I suggest correcting this sentence in English language editing, write in lower cases the cardiovascular.

Line 53: correct the sentence in English language editing, not “closed to” but “close to”, rather say large than huge

Line 58: I suggest replacing “high” with “very”.

Line 68: the sentence is not clear in the meaning, did the authors mean that all-trans RA induces weight gain and increased plasma TG and cholesterol?

Line 70: correct the sentence, I suggest: “In addition to clinical observation, animal experimental studies

had also demonstrated that plasma HDL-C levels decreased in obese rats fed with VA [21].” Were they fed with higher levels of VA?

Line 75: the sentence is not clear, what is The hypidemic effect? What is the source of this statement?

Line 77: define HFD meaning

Lines 79-89: those sentences should be corrected, I suggest clearly writing the conclusion from the above-mentioned animal research, and stating the aims of this research by avoiding the “we” and to put the hypothesis of the research.

Materials and Methods

Line 92: write “this study” rather than our.

Line 96: write the number of surveyed children, what were the inclusion and exclusion components of the survey, were all children that were physically examined, venipunctured, and questioned included in this study or were all, had they all their data or there were missing data for some variables or questions in this survey? Define the term “mixed hyperlipidemia”.

Line 103: write “the questionnaire survey”

Line 104: write this sentence correctly, I suggest: The working groups from the nation and provincial working groups were responsible for the quality control of the survey.

Line 115-116: avoid “we”. Define the age of children that fulfilled the questionnaire by themselves and those who fulfilled with the help of their guardians. Were they previously instructed how to fulfill the FFQ and by who?

Line 118: avoid “we”, correct the sentence that the participants were classified into three groups based on WHO criteria and Chinese paediatric textbooks.

Line 122: avoid “we”, write in lower cases.

Line 128-138: correct the sentences in English editing, the variables don’t use the tests but the test was used to test the variables. Write the adjusted variables and confounders relevant for the included statistical tests, also the logistic regression independent and dependent variables.

Results

In tables and figures, titles write all details and information about as they should stand alone, for example, “Table 1. The characteristics of 3551 children from Shandong province, China according to their vitamin A status”, write the p-values in tree decimal places, uniformly, as well all results show in one or two decimal places, but all uniformly. In the table, subheadings write the methods that were used for statistical analysis between groups (ANOVA?). I suggest presenting the study results differently. Table 1 shows the demographic and anthropometric characteristics and serum characteristics (lipid profile/status, show serum vitamin A values, etc.). Table 3 shows dietary factors, then include energy intake (MJ/day), macronutrients (proteins, total fats, SFA, MUFA, PUFA, carbohydrates –all in g/day and as a proportion of total energy intake-%), cholesterol, vitamin A, D, E, beta-carotene, dietary retinol, and if it is possible, the average intake of food groups (g/day), especially those relevant for vitamin A content (beta-carotene/retinol also). Perhaps the presented dietary intake of selected variables of participants' diet could explain their serum variables results as well as group differences.

Line 156: the sentence cannot start with “and”, mentioned the source of those results (Table 1), mention that those results were compared to the other groups, and uniformly write the p-value as shown in the table. 

174-185: the results of logistic regression, if adjusted, their adjustment variables should be explained in methods, and then be shown in tables. There is no need to write again the results from the table, mention the differences and statistically meaningful results.

Line 189-195: the authors should explain in the methods how they did their models (Model 1 to 4), not to describe them in table subheadings. This can be uniformly explained in the statistical analysis. It is confusing and voluminous for the table that is showing.

Discussion

Line 219: it cannot be said that the nutritional status is not optimistically, rather it should be stated that the study results showed that among Chinese children is present serum vitamin A deficiency, almost in a quarter of surveyed children, and then discuss the results meaning for Chinese children or others. At the start of the Discussion, the authors should answer their study aims, and see if there is an explanation for provided results (associations, etc.), to explain the hypothesis results (confirmed or rejected).

I suggest writing a new discussion if the authors accept to show some more results as suggested above (more dietary variables). It is very interesting that the authors found the association with the high CRP and serum vitamin A deficiency, in the context of immunity. I suggest mentioning this in the Introduction. In addition, if the authors are willing and in possibility, they could do the diet analysis for the inflammatory potential with the Dietary inflammatory index, DII® (Shivappa, N.; Steck, S.E.; Hurley, T.G.; et al. Designing and developing a literature-derived, population-based dietary inflammatory index. Public Health Nutr 2014, 17, 1689-1696.). The provided results of DII can be associated with anthropometrics, CRP, serum lipids, hyperTG, hyperlipidaemia, serum vitamin A and D, etc. Those results can be shown in the presented groups. It could be shown in the Supplementary Materials. When discussing, avoid “we”, write in passive voice.

The study conclusion should highlight the main associations explaining their meaning, not to mention again the variables. For example, they could conclude that this study found a positive association with hyperlipidaemia, but was influenced by age. The authors then do not mention what were the age groups.  For example, Younger children had lower, but positive associations when compared with older children which also had positive associations. Here, the authors should mention what the study findings mean for the paediatric societies.

Author Response

Please review the attachment.

Reviewer 2 Report

Thank you for your contribution to this journal.

1. You described that, 'voluntarily', attendance. Such a young and little children are able to attend voluntarily?

2. Many years ago, vitamin A deficiency is very popular issue in China. Is this study is worthy to worry about to high serum A level?

3. What is the reason to devide the A level into 3 groups such as <0.7, 0.7-1.05 , >1.05 , not with tertile, is the A level showed normal destribution?

4. OR & 95%CI in table 3, are very difficult to read. I was so confused to recognize. Neet to be written again.

5. What is the main food sourcse of vitamin A in China?

Author Response

Please review the attachment
